# The Nutrition Transition and the Double Burden of Malnutrition in Sub-Saharan African Countries: How Do These Countries Compare with the Recommended LANCET COMMISSION Global Diet?

**DOI:** 10.3390/ijerph192416791

**Published:** 2022-12-14

**Authors:** Johanna H. Nel, Nelia P. Steyn

**Affiliations:** 1Department of Logistics, Stellenbosch University, Stellenbosch 7600, South Africa; 2Department of Human Biology, University of Cape Town, UCT Medical Campus, Anzio Road, Anatomy Building, Observatory, Cape Town 7925, South Africa

**Keywords:** food security, diet, malnutrition, double burden, sub-Saharan Africa, dietary patterns, climate change, non-communicable diseases

## Abstract

Background: Over the last two decades, many sub-Saharan African (SSA) countries have undergone dietary and nutrition transitions fuelled by rapid urbanisation, economic development, and globalisation. The aim of the current study was to examine outcomes of the nutrition transition and the epidemiologic transition in SSA countries in terms of food intake, health, and socioeconomic and development factors. Methods: Food balance sheet data—specifically, per capita energy intake per day and per capita gram intake per day—from the CountrySTAT framework of the Food and Agricultural Organization (FAO) were analysed for major food commodities. Additionally, selected health and development indicators supplied by UNICEF, the WHO and the World Bank were analysed. Results: Four dietary patterns emerged. The diet of the southern African/island cluster (South Africa, Mauritius, Eswatini, Namibia, Cabo Verde, and the outlier Seychelles) resembles a Westernised diet, with median values high on sugar/sweeteners, alcohol, meat, animal fats, eggs, and dairy. On the other hand, the diet of countries in the other three clusters appears to be more traditional, with countries in the desert/semi-arid cluster consuming more cereals and pulses/tree nuts, countries in the tropical coastal cluster consuming more fish and vegetable oils, and countries in the equatorial cluster consuming more starchy roots and fruit and vegetables. The resulting median values of health indicators also indicate a higher prevalence of non-communicable diseases in the southern African/island cluster, whereas stunting and anaemia are higher in the other three clusters. Conclusions: SSA countries are in different stages of the nutrition transition. By superimposing clusters generated using macronutrient intake values on a map of the climatic regions in Africa, one can clearly see the importance of climate on the availability of food and food intake patterns. Climate change presents a great challenge to healthy eating, as the link between climate regions and diets is illustrated.

## 1. Introduction

Globally, we are challenged by numerous obstacles to a safe, healthy, and sustainable food supply. Poor agricultural practices, internal conflict and war, globalisation, urbanisation, climate change, poor governance, low socioeconomic development, and population growth are among the major drivers of poor food security in middle- and low-income countries [1,2,3]. Laar and colleagues have argued that over the last two decades, many African countries have undergone dietary and nutrition transitions fuelled by rapid urbanisation, development, and globalisation. These changes have altered dietary behaviours, including how food is acquired, processed, and consumed [4]. Dietary patterns have emerged which are associated with a complex burden of malnutrition. On the one hand, undernutrition is still highly prevalent, especially in children. On the other hand, obesity and other diet-related non-communicable diseases such as diabetes, certain cancers, hypertension, and coronary heart disease have become prevalent. When a country has a high prevalence of both undernutrition, such as stunting, and overweight individuals and obesity, we refer to the double burden of malnutrition (DBM) [5,6].

First described by Popkin, the nutrition transition is the outcome of major shifts in the diet towards increased refined carbohydrates, added sweeteners, edible oils, and animal-source foods, as well as reduced intake of fruit, vegetables, and legumes [7]. This pattern has been associated with several changes including economic growth, fast urbanisation, and an increase of highly processed foods.

Now termed *ultra-processed foods*, these are produced by: “processes enabling the manufacture of foods including the fractioning of whole foods into substances, chemical modifications of these substances, assembly of unmodified and modified food substances, frequent use of cosmetic additives and sophisticated packaging” [8]. Examples are soft drinks, savoury snacks, many packaged breads, confectionary, and sweet and savoury biscuits [9,10]. Two recent reviews have shown that ultra-processed foods have numerous adverse health outcomes. These included overweight, obesity, type 2 diabetes, cardiometabolic risks, cardiovascular diseases, irritable bowel syndrome, cancer, depression, and all-cause mortality [11,12].

The nutrition transition has also been accompanied by changes in the health of the population (epidemiological transition), with a shift occurring from mainly infectious diseases and undernutrition to one of obesity and non-communicable diseases (NCDs) including cardiovascular diseases (CVD), certain cancers, and type 2 diabetes. According to Cowan et al., 87% of premature NCD mortality occurs in low- and middle-income countries [13]. However, diet alone is not responsible for the increase in NCDs, since physical inactivity is also a major contributing factor to the global pandemic of NCDs. Low levels of physical activity are seen in an increasing proportion of populations across low- and middle-income countries and high-income countries alike [14]. This epidemiological shift was first observed mainly in developed countries such as Britain, USA, and Europe. However, in recent decades the transition has also shifted to low and middle-income countries including those in sub-Saharan Africa (SSA). The rapid rise in cardiovascular burden in most of the low- and middle-income countries is due to socioeconomic changes, acquisition of lifestyle-related risk factors, and increase in lifespan [15].

Sub-Sahara’s 143 cities generate a combined USD 0.5 trillion. This represents 50 percent of the region’s gross domestic product (GDP) [16]. Urban centres sustain economic growth, are often considered the future of prosperity in the developing world, and play an important role in fighting poverty. However, strategic urbanisation (planning for increased population) is vital and is dependent on numerous role players, including economic policymakers, the private sector, city governments, development, and urban planners. Bringing together the different stakeholders in urbanisation management presents a vitally important opportunity for economic growth in a region which is undergoing an immense demographic shift [16].

Saghir and Santoro [16] from the Centre for Strategic and International Studies have indicated that urbanisation is one of the most important transformations that SSA countries will undergo in this century. The consequences of urbanisation can interact with other development issues such as policy, migration, and climate change. If not managed holistically, it is predicted that the inflow of informal settlers will continue, with an additional 560 million people expected to move into SSA cities by 2040.

Likewise, Dia and Beaudelaire [17] have pointed out that in recent years, Africa has developed much faster than other countries in the world. This is thought to be due to factors above initial endowments and favourable geographic conditions. Some of the internal drivers include the strengthening of policies, improvement of government—making it effective and efficient, favourable conditions for agricultural activity, and the emergence of a middle class. Regarding external influences, soaring prices and capital inflow are believed to be the main drivers of economic growth. According to the United Nations, projections indicate that SSA will become increasingly urban between 2025 and 2035, with 1.2 billion people expected to live in cities by 2050 [17].

According to the World Health Organisation (WHO) a healthy diet is one which includes the following guidelines [18]. Firstly, achieving energy balance and a healthy weight; limiting energy from total fats and giving preference to unsaturated fats versus saturated fats; increasing fruit, vegetables, legumes, whole grains, and nuts; limiting the intake of free sugars and salt, and ensuring that salt is iodised.

More recently, the Lancet Commission developed a diet which would, in theory, be a healthy diet for the global population [19]. This Commission, which comprised more than 35 international renowned nutrition and environmental scientists, developed the Lancet diet for a global population which is increasing rapidly, and which would provide a sustainable healthy diet for all, taking the environment and climate into consideration. Their universal healthy reference diet largely consists of vegetables, fruit, whole grains, legumes, nuts, and unsaturated oils; includes a low to moderate amount of seafood and poultry; and includes no, or a low quantity of, red meat, processed meat, added sugar, refined grains, and starchy vegetables. It is, in fact, very similar to the WHO recommendations [18], except that the Lancet Commission went a step further and made specific recommendations regarding the daily quantities of food groups which should ideally be consumed in a diet comprising 2500 kcal for the average woman, and which would meet basic human nutrient requirements. There are, however, numerous criticisms of the global diet. These include the fact that it is not affordable for millions [20,21], that it is not sustainable [22], that it is based on inaccurate modelling of data [23], and that it is not water-wise regarding tree nuts and groundnuts, for example [24].

The aim of the current study was to examine outcomes of the nutrition transition and the epidemiologic transition in SSA countries in terms of food intake, health, and related factors. SSA comprises mostly low-income countries and is expected to be most vulnerable to the nutrition transition and its epidemiological outcomes in terms of NCDs. Furthermore, we compared the food groups consumed in the SSA countries with two Western high-income countries, namely, USA and UK, and we compared the per capita intakes with the Lancet global diet recommended intakes.

## 2. Materials and Methods

### 2.1. Food Intake Data and Methodology

Three different groups of data are analysed to describe food intake, health, and development, as well as associations between these. The Food and Agricultural Organisation (FAO) compile the CountrySTAT framework [25] which provides statistics on the existing national system of food and agriculture for 181 countries. Of specific interest are the food balance sheets (FBSs) [26].

Food balance sheets provide essential information on a country’s food system through three components: Per capita values for the supply of all food commodities in terms of kilograms per person per year and in terms of calories, protein, and fat intakes per person per day. The food balance sheets further provide the domestic food supply of each country in terms of imports, production, and stock changes. Lastly, they present food utilisation including feed, seed, waste, export, and other uses. The food balance sheets provide the total quantity of foods produced in a country based on production and allowing for imports and exports over a 1-year period. The per capita supply of each such food item available for human consumption is obtained by dividing the respective quantity available domestically and dividing it by the population consuming it [25,27].

The main food item categories and their codes, as described in the food balance sheets [26] are listed in Table 1. Of specific interest are the per capita intake, in terms of g/capita/day, and the calorie intake, in terms of kcal/capita/day. These values will be analysed per country based on the food balance sheets for 2019.

Considering the food intake categories in Table 1, the first step was to cluster countries according to their eating patterns, using a combination of factor analysis and cluster analysis. Exploratory factor analysis (EFA) was used to identify the latent structure underlying a set of variables, or food intake categories, in this case [28]. Factor scores can then be obtained for each country on each factor and can thus be interpreted as numerical values that represent a country’s relative standing on each factor. The dimension of the data is therefore reduced, and clustering countries using the factor scores can reveal a meaningful structure among countries that are based on combinations of food intake categories, whereas clustering on the original food intake categories can produce clusters regardless of the actual existence of any structure. Therefore, dietary patterns are generated using EFA, and the countries are grouped into clusters according to their dietary patterns [29].

The underlying pattern of the 2019 kcal/capita/day intake of the 12 food intake categories, as described in Table 1, was estimated. Exploratory factor analysis was carried out with maximum likelihood parameter estimates on standardised variables. Initially, no rotation was specified. Three factors were retained. A new dataset was created which contains the factor scores for each observation. Correlations between these factor scores revealed that the maximum absolute correlation between any two factors was 0.24596, and all associated *p*-values were greater than 0.1. As the correlations between factor scores were low, orthogonal varimax rotation was applied. The three factors retained in the analysis were based on retaining factors that accounted for more than 10% of the common variance as well as interpretability. The Kaiser–Meyer-Olkin (KMO) test was also considered to examine adequacy [30]. KMO values above 0.5 are generally accepted as indicating the adequacy of the sample for factor analysis. A cut-off KMO value of 0.55 was used, and only pulses, tree nuts, and vegetable oils had KMO values between 0.55 and 0.60. Foods with a factor loading coefficient > 0.30 were considered to provide the factor with a descriptive name. Estimated factor scores were calculated, which represent each factor for all SSA countries.

The 3 factor scores were used as input in Ward’s analysis to cluster SSA countries [31]. A pseudo-t^2^ criterium was used to set the chosen number of clusters to 4. After removing Seychelles (outlier) from the list (which clustered initially with South Africa and Mauritius), a more distinct cluster pattern was obtained.

### 2.2. Data on Burden of Malnutrition, NCD Health, and Development Indicators

To investigate malnutrition estimates for children under 5 years of age, the UNICEF/WHO/World Bank data were used [32,33]. The data is derived from most recent Demographic and Health Surveys (DHSs), Multiple Indicator Cluster Surveys (MICSs), and other major surveys and were reanalysed by external partners. The prevalence of stunting is the percentage of under-5-year-old children in each country falling below minus 2 standard deviations (moderate and severe) from the median height-for-age of the reference population. Overweight prevalence is provided as the percentage of children under 5 years of age falling above 2 standard deviations (moderate and severe) from the median weight-for-height of the reference population, and the prevalence of concurrent stunting and overweight [34] are those children under 5 years of age experiencing simultaneous stunting and overweight. Data on women’s overweightness and obesity was obtained from the Global Nutrition Report [35]. Overweightness/obesity prevalence is based on age-standardised modelled estimates for females aged 18 years and older, using the WHO standard population. Data on anaemia in children under 5 and women of reproductive age were obtained from the WHO [36]. Hypertension data of males and females, 2019, by country, age-standardised (30–79 years), were obtained [37], as well as data on diabetes based on aged-standardised modelled estimates for adult females 18 years and older using the WHO standard population [38]. Data on blood cholesterol were also included [39].

Some development indicators from the World Bank 2020 [40] were also included, such as the gross domestic product (GDP) in US dollars, the Gini index, % population 65 years and older, % urban population, and % population living in informal settlements. The birth rates for each SSA country, representing the total births per women, were also sourced from the World Bank [41]. The Gini index measures the extent to which the distribution of income (or, in some cases, consumption expenditure) among individuals or households within an economy deviates from a perfectly equal distribution. A Gini index of 0 represents perfect equality, while an index of 100 implies perfect inequality. The data for the Gini index for the different countries vary from the oldest, in the Central African Republic, from 2008; in Republic of the Congo, from 2011; and in the rest of the SSA, from 2012 onwards [40].

### 2.3. Associations between Food Clusters and Health and Development Indicators

Associations between the resulting food intake clusters and malnutrition, and NCD health characteristics, namely, child (<5 years, 2021) stunting [32], child (<5 years, 2021) overweightness [33], concurrent stunting and overweightness (children < 5 years, 2021) [34], standardised overweight status of females 18 years and older (2017) [35], child (<5 years, 2019) anaemia [36], anaemia among women of reproductive age (15–49 years) as of 2019 [36], hypertension prevalence for males and females (2019 estimates, age-standardised, aged 30–79 years) [37], type 2 diabetes in women [38], adult females 18 years and older (2019 estimates), and blood cholesterol, were carried out [39]. Associations between the dietary pattern clusters were also carried out with development indicators from the World Bank, 2020 [40,41].

The median and interquartile range per cluster for each of these variables were calculated, and the Kruskal–Wallis test was used to assess associations between the 4 clusters and health and development indicators. The Bonferroni test was used to test for differences between cluster groups.

Furthermore, FBS data [26] were used to calculate the following for each SSA country in g/capita/day: starchy roots, cereals, vegetable oils, meat, animal fat, sugar, fruit and vegetables, dairy, fish and seafood, eggs, and alcoholic beverages. Per capita intakes for the UK and USA are also shown in order to provide a perspective of comparison with developed countries. The DBM for SSA countries is shown as a Venn diagram to illustrate countries in the highest tertiles for overweight/obesity in women and stunting in children under 5 years of age. The epidemiological transition is shown by a Venn diagram which presents the highest tertiles of type 2 diabetes, hypertension, overweightness/obesity of women, and child overweight/obesity prevalence of each SSA country. Lastly the nutrition transition of SSA countries is presented as a Venn diagram showing the highest tertiles of sugar, vegetable oil, and meat, and lowest intake of fruit and vegetables.

## 3. Results

The results of factor analyses of the food items are presented in Table 2. The food intakes of the 12 food intake categories load onto three factors, which can be described as a typical Westernised diet (Factor 1), a combination of a traditional and Westernised diet with the focus on cereals, but low on fruit and vegetables and starchy roots (Factor 2), and a combination of a traditional and Westernised diet with the focus on eggs, fish, and vegetable oils but low on pulses and tree nuts (Factor 3). A traditional diet is defined as one containing plant and animal foods harvested from the local environment; these foods are often called “country foods” to reflect their origin from the land [42]. Over recent decades, the diets of children and adolescents in Western countries reveal increases in the intake of energy-dense, heavily processed foods rich in saturated fat, salt, and sugars. This is referred to as a Western diet [43].

A set of three factor scores were created for each SSA country, and the SSA countries were subsequently grouped into four clusters based on these factor scores (Table 3). The clusters represent countries in desert or semi-arid areas (Cluster 1); countries in tropical savannah or monsoon regions, mostly coastal countries (Cluster 2); tropical or sub-tropical countries in the equatorial region (Cluster 3); and mostly southern African countries combined with developed islands (Cluster 4).

Figure 1a,b compare the regions generated by clustering food intake (kcal/capita/day) data with the Köppen climate regions [44]. Many countries have more than one climate region and it is not possible to obtain a one-to-one comparison for all the countries, but almost all countries overlap mainly with the climate region as described in the cluster.

Table 4 compares intake of the main food items, in g/capita/day, for the four clusters. The desert/semi-arid cluster (Cluster 1) clusters highest on cereals and pulses/tree nuts, and lowest on starchy roots, fruit, and vegetables. The tropical coastal cluster (Cluster 2) clusters highest on vegetable oils, fish, and seafood and lowest on alcoholic beverages, meat, animal fat, and milk. These countries are mainly in coastal regions. The equatorial cluster (Cluster 3) clusters highest on starchy roots, fruit, and vegetables and lowest on total energy intake, cereals, sugar/sweeteners, vegetable oils, and eggs. These countries are mainly around the equator and are subjected to high temperatures and rainfall. The southern African and island cluster (Cluster 4) also clusters high on total energy intake, sugar/sweeteners, alcohol, meat, animal fat, eggs, and dairy and lowest on pulses/tree nuts.

Countries in the tropical coastal and equatorial clusters (Cluster 2 and 3) have the lowest median calorie intakes, of between 2300 and 4000 calories per capita per day, while countries in the desert/semi-arid regions and the southern African/island regions (Clusters 1 and 4) have the highest median energy intakes, of around 2500 calories (Table 4).

There are significant differences between countries regarding cereal intakes with those in the desert/semi-arid cluster (Cluster 1) having the highest median intake (477 g/capita/day) and those in the equatorial cluster (Cluster 3) having the lowest median intake (257 g/capita/day). Countries in the southern African/island region (Cluster 4) have the highest median sugar and sweeteners, meat, animal fats, eggs, dairy, and alcoholic beverages. They also have the lowest intake of starchy vegetables and pulses, and their fruit and vegetable intake places them below the 400 g recommended by the World Health Organisation [46]. Their profile places them in the highest category of the nutrition transition while the equatorial cluster (Cluster 3) countries have the lowest median intake.

The Bonferroni test shows that there is a significantly higher intake regarding sugar and sweeteners in Cluster 4 in comparison with the other three clusters. The same applies to meat, animal fats, eggs, and dairy products.

In Figure 2a it is shown that there are 6 countries with more than 500 g/capita/day cereal intake (Mauritania, Guinea, Niger, Burkina Faso, Senegal, and Mali). The equatorial cluster (Cluster 3) has the lowest median intake of cereals, of 257 g/capita/day. It is interesting to note that the USA and UK fall within the bottom third of countries regarding cereal intakes. The equatorial cluster (Cluster 3), on the other hand, has the highest intake of starchy roots, of 736 g/capita/day (Figure 2b). There are three countries (Cote d’Ivoire, DRC, and Ghana) which have more than 800 g/capita/day of starchy roots. The UK and USA have less than 200 g/capita per day. The desert/semi-arid cluster (Cluster 1) has the highest per capita intake of cereals and the equatorial cluster (Cluster 3) the lowest, at 477 g/capita/day and 257 g/capita/day, respectively. Most countries have a sugar intake of less than 50 g/capita/day (Figure 2c). However, seven countries have intakes greater than 100 g/day, with Eswatini reaching almost 200 g/capita/day. The UK and USA also fall in the highest group of consumers, with over 100 g/capita/day. The USA has a per capita intake of 181 g/day. Overall, the tropical coastal and equatorial clusters (Clusters 2 and 3) have the lowest median sugar intake, of 33 g/capita/day. Those in the desert/semi-arid cluster (Cluster 1) reach a median of 66 g/capita/day, and in the southern African/island cluster (Cluster 4), this reaches 110 g/capita/day. Six countries (Ethiopia, Tanzania, Cameroon, Burundi, Rwanda, and Niger) have a per capita intake of pulses and tree nuts greater than 60 g per day, although the vast majority of countries have less than 30 g/day (Figure 2d). The UK and USA also have a low per capita intake of 12 g/day and 18 g/day, respectively.

Figure 3 presents per capita intake of animal products in SSA. Three countries (Mauritius, Seychelles, and South Africa) have a per capita meat intake greater than 150 g/day (Figure 3a). However, the majority of countries have a per capita intake less than 50 g/day. The UK and USA have very high intakes, at 216 g/day and 352 g/day, respectively. Clusters 1, 2, and 3 have median meat intakes of 37 g/day, 42 g/day, and 45 g/day, respectively, while the southern African/island cluster (Cluster 4) has a median intake of 90 g/day. Eight countries (Mauritius, Ghana, Sierra Leone, Gambia, Congo, São Tomé and Príncipe, Gabon, and Seychelles) have per capita intakes of fish/seafood greater than 60 g/day (Figure 3b). These are islands, have coastal lands, or are countries with vast lakes and rivers. Most countries have a per capita fish/seafood intake of less than 20 g/day; the desert/semi-arid cluster (Cluster 1) has a median intake of 10 g/day. The UK and USA have per capita intakes of 51 g/day and 61 g/day, respectively. Mauritius, Kenya, and Botswana have a per capita milk intake greater than 200 g per day; Cluster 4 countries have a median intake of 110 g/day (Figure 3c). By contrast, the UK and USA have per capita intakes of 575 g/day and 632 g/day, respectively. Most SSA countries lie below 50 g/day in per capita milk intake. The intake of eggs overall is low (Figure 3d): a per capita intake of less than 10 g/day. The UK and USA have per capita intakes of 31 g/day and 45 g/day, respectively.

All SSA countries have a per capita animal fat intake of less than 10 g/day (Figure 4a). The UK and USA have per capita intakes of 10 g/day and 12 g/day, respectively. A total of 14 SSA countries have a per capita vegetable oil intake greater than 30 g/day, while a few, such as Burundi, Madagascar, and Ethiopia, have intakes less than 10 g/day (Figure 4b). The UK and USA have per capita vegetable oil intakes of 37 g/day and 55 g/day. A total of 10 countries have a fruit and vegetable intake greater than 400 g/day: Senegal, Gabon, Mali, Seychelles, Uganda, Rwanda, Cameroon, Ghana, São Tomé and Príncipe, and Malawi (Figure 4c). A total of 20 countries have a per capita intake of less than 200 g/day. The UK and USA have fruit and vegetable per capita intakes of 433 g/day and 566 g/day, respectively. Six countries have an alcohol intake greater 200 g/day, while the majority have an intake of 100 g or less. The UK and USA have per capita alcohol intakes of 243 g/day and 248 g/day, respectively (Figure 4d).

Figure 5 presents the calories per capita intakes of all SSA countries and the UK and USA. Mauritius, Ghana, and Seychelles are the only three SSA countries reaching 3000 kcal/day. Zimbabwe, Burundi, CAR, Madagascar, and the DRC possess the lowest per capita intakes: less than 2000 kcal/day. In total, 24 SSA countries have a per capita intake of less than 2500 kcal/day. Three SSA countries have per capita intakes of 3000 kcal/day (Mauritius, Ghana, and Seychelles). The UK and USA have the highest per capita intakes, of 3395 kcal/day and 3862 kcal/day, respectively.

Table 5 presents data on the weight status and health of the populations in the various clusters. Cluster 4 is markedly different from the other clusters. It has the lowest median child stunting, at 22.1%, while the tropical coastal and equatorial clusters (Clusters 2 and 3) have the highest median, around 30%. Cluster 4 has the highest median child overweightness, at 7.8%, while the median of the other clusters range between 2.1 and 4.5%. It also has the highest median mother overweightness, at 51.9%, while the other clusters range between 34.9 to 37%. Additionally, it has the highest median prevalence of hypertension (38.1%) in contrast with the other clusters, which vary between 35.6% and 36.7%. The median of type 2 diabetes is significantly higher in women in Cluster 4, at 11.3% (*p* < 0.01), compared with other clusters. The median of anaemia is significantly lower in the desert/semi-arid cluster (Cluster 1) in both children and their mothers, at 44.1% and 25.2%, respectively (*p* < 0.01). The Bonferroni test shows that there are significantly higher values in Cluster 4 in comparison with the other three clusters regarding child overweightness, females overweight, and females with type 2 diabetes. Cluster 4 has significantly lower values for child and female anaemia.

Table 6 presents development indicators for the four clusters. The median GDP value for Cluster 4 is nearly 5 times higher than for the other clusters (*p* < 0.05), as well as the percent of people over 65 years of age (4.8%) (*p* < 0.01). Although not significant, Cluster 4 has the highest median percentage of urban dwellers (52%), compared with 35.4% to 44.4% in the other clusters. The median percent of those living in informal settlements is lowest in Cluster 4 (32.1%), compared with 48.3 to 57.1% in the other clusters. Cluster 4 also presents a significantly lower median birth rate, of 2.4, compared to nearly double that of the other clusters. The Bonferroni test shows that Cluster 4 has significantly higher values for GDP, % over 65 years, and Gini coefficient. Cluster 4 has significantly lower annual population growth and birth rate than the other clusters.

Figure 6 shows those SSA countries having the highest tertile of child stunting and overweightness in children under five. In total, 10 countries lie in the top tertile for stunting and for overweightness. Four countries lie in the top tertile for both child stunting and overweightness: Djibouti, Lesotho, Mozambique, and Rwanda. In Figure 7, the DBM is illustrated. Djibouti and Lesotho show the highest DBM, with the top tertiles for both child stunting and women overweight at the national level.

Figure 8 presents data on countries illustrating evidence of the epidemiological transition. Three countries, Lesotho, Eswatini, and Botswana, show four risk indicators; namely, they lie in the highest tertile for women and children being overweight, women with type 2 diabetes, and individuals with hypertension. Countries with three of the risk factors include Comoros, Gabon, Seychelles, and South Africa.

There are two countries which show high evidence of the nutrition transition, namely, Mauritania and South Africa (Figure 9). They have a high intake of meat, oil, and sugar, and a low intake of fruit and vegetables. There are two countries which are high in oil and sugar intake and low in fruit and vegetables: Botswana and Zimbabwe. Furthermore, Mauritius also has three risk factors: a high meat, sugar, and vegetable oil intake.

Botswana and South Africa have the highest GDP, percent urbanisation, percent of population 65 years and older and the highest Gini coefficient (Figure 10). Eswatini and Namibia have a high GDP, high % population over 65 years and a high Gini coefficient. Djibouti, Gabon, and Seychelles have all the indicators except for a high Gini coefficient. Angola and Congo also have all the indicators in the top tertile except for a high population over 65 years.

## 4. Discussion

In summary, countries in the southern African/island cluster (Cluster 4) have the highest median intake of sugar and sweeteners, meat, animal fats, eggs, dairy, and alcoholic beverages. They also have the lowest intake of starchy vegetables and pulses, and their fruit and vegetable intake places them below the 400 g recommended by the WHO [46]. Their profile places them in the highest category of the nutrition transition, while the equatorial cluster (Cluster 3) countries have the lowest median intake of cereals, animal products, vegetable oils, and sugar and sweeteners, and the highest intake of starchy roots and fruit and vegetables, most resembling the more traditional African diet.

Regarding the DBM, it is shown that Djibouti and Lesotho have both the highest child stunting and percent of women overweight/obese in the highest tertile. In total, 12 countries are found in the highest tertile for stunting, and 11 for women who are overweight/obese. Three countries (Lesotho, Eswatini, and Botswana) show four risk indicators of the epidemiological transition; namely, they lie in the highest tertile for women and children being overweight, women with type 2 diabetes, and individuals with hypertension. Countries with three of the epidemiological risk factors include Comoros, Gabon, Seychelles, and South Africa. There are two countries which show four risk indicators of the nutrition transition, namely, Mauritania and South Africa. They have a high intake of meat, oil, and sugar and a low intake of fruit and vegetables. There are a number of countries which have three risk indicators of the nutrition transition: Botswana, Zimbabwe, Mauritius, and Namibia. However, it needs to be considered that, when compared with the UK and USA, some of these risk indicators are still relatively low.

The findings on the nutrition and epidemiological transitions can be explained to some extent by selected development indicators. Botswana and South Africa are in the highest tertile for all four development indicators (highest GDP, highest Gini coefficient, highest percent urbanisation, and highest percent of population 65 years and older). Several countries have three of the four development indicators; namely, Eswatini and Namibia have a high GDP, percent over 65 years of age, and a high Gini coefficient. Djibouti, Gabon, and Seychelles also have all the indicators except for a high Gini coefficient. Angola and Congo also have all the indicators in the top tertile except for a population percent greater than 65 years. If we examine the countries in the high nutrition transition, high epidemiological transition, with high development indicators, Botswana and South Africa appear to have at least three indicators in each.

Overall, when examining SSA countries we can notice certain trends which are not indicative of a healthy diet as recommended by the WHO Global Strategy of Diet, Physical Activity and Health, which aims for individuals to achieve energy balance and a healthy weight; limit energy from total fats and grant preference to unsaturated fats versus saturated fats; increase fruit and vegetables, legumes, whole grains and nuts; limit the intake of free sugars and salt, and ensure salt is iodised [18].

The Global Burden of Disease (GBD) study outlined the morbidity and mortality that result from a poor dietary intake in 195 countries [47]. It was noted that the highest rates of mortality and disability-adjusted life years (DALYS) related to diet were in low- and middle-income countries. Risk factors associated with the highest rate of mortality in descending order were: a diet high in sodium, low in whole grains, low in fruit, low in nuts and seeds, low in vegetables, low in fish/seafood (omega-3-rich fatty acids), low in fibre, low in polyunsaturated fats, low in legumes, high in trans fats, low in calcium, high in sugar-sweetened beverages, high in processed meat, low in milk, and high in red meat. In 25–49-year-olds, high systolic blood pressure, high body mass index, high fasting plasma glucose, and high LDL cholesterol are the second, third, sixth, and seventh leading disease risk factors, respectively [47].

When one examines the calorie distribution of SSA countries, it is noticeable that there are five countries who have a per capita kcal/day less than 2000 kcal, and 19 with intakes less than 2500 kcal/day. The Institute of Medicine indicates that the average male 19 years and older who is moderately active requires 3067 kcal/day, and the average women 19 years or older requires 2403 kcal/day [48]. Several countries in the tropical coastal and equatorial clusters (Clusters 2 and 3) have low intakes in comparison to the other two clusters and in comparison with the recommendations of the Institute of Medicine [48]. This is indicative of food insecurity being present in those countries. As mentioned earlier, food insecurity may be the outcome of numerous factors. In SSA, some of the most prominent factors are drought, conflict, urbanisation, high birth rates, and poor governance. Unfortunately, the frequency of droughts is expected to increase in coming decades due to climate change. Sub-Saharan Africa is the most vulnerable region to climate change. This is due to long dry seasons, low rainfall, low adaptive capacity, and high levels of poverty [49].

Droughts affect more people than any other natural hazard [49], especially the poor, who are more vulnerable, and this is especially evident in SSA, which has the highest global food insecurity [50,51]. Droughts are among the main threats affecting agricultural productivity and, in turn, household food security and health [52,53]. According to Dumenu and Obeng [54], nearly two in three people who live in rural areas in SSA rely solely on rain-fed agriculture as a source of income and to sustain their livelihoods. Additionally, food security in SSA is compounded by extremely rapid population growth which intensifies poverty and lack of economic opportunities [3].

Armed conflict also affects people’s food security. Africa is particularly vulnerable since it buys much of its wheat, maize, sunflower oil and fertilisers from Ukraine and Russia. Currently, due to the Ukrainian war this supply is seriously hampered, which may lead to widespread famine in North Africa and selected SSA countries within the coming years. Many of these countries have recently dealt with drought, the COVID-19 pandemic, and extreme Islamism and have yet to recover [55].

In SSA, urbanisation is increasing at a rate of 4.1% compared with the global figure of 2%. In the next 30 years, it is estimated that urban dwellers will outweigh rural ones for the first time. However, much of the urbanisation is of an unplanned nature and results in inadequate infrastructure such as inadequate housing, unsafe drinking water, poor sanitation, and poor access to health care. These factors will make the residents ill-equipped to deal with the risks they may be exposed to, including food insecurity [16,56].

The traditional African diet is high in cereals and/or starchy roots, and generally supplies more than 400 g/day, or 60–75% of energy intake [57]. This contrasts with the more Westernised diet of the UK and USA, in which cereals and/or starchy roots generally supply less than 400 g/day, amounting to 45–65% of the recommended energy intake per day [47]. It is also known that the variety of cereals consumed, including maize, wheat, rice, barley, rye, oats, millet, and sorghum, are generally preferred in the refined form, which limits the intake of fibre and micronutrients [58]. While many cereals were hand-ground/milled in the past, today many are processed at mills and highly refined. The Lancet global diet [19] recommends an intake of whole grains of 232 g/day and 50 g/day of potatoes or cassava, with an intake contributing to less than 60% of total energy intake. Currently, only 11 SSA countries lie withing the grain recommendation and 5 countries fall within the recommendation for tubers

The overall per capita intakes of pulses (beans, lentils, and peas) and tree nuts were low in the majority of SSA countries, amounting to less than 20 g per day, with only 7 countries possessing a per capita intake of at least 50 g/day. A high intake of legumes and nuts are part of a traditional African diet, and it is disappointing that the intakes in SSA are so low, and more in line with a Westernised diet as shown by the UK and USA intakes. Since legumes are a good source of relatively inexpensive protein, folate, fibre, iron, phosphorus, and mono- and polyunsaturated fatty acids, and have a low glycaemic index, their intake is highly recommended [59]. They are also believed to prevent the development of certain NCDs. A recent meta-analysis showed that the consumption of four weekly 100 g servings of legumes was associated with a 14% lower risk of ischemic heart disease [60]. About three cups per week of legumes are recommended by the US Dietary Guidelines, and four to five half-cup servings per week in the DASH diet [61]. Despite being a rich source of iron, the bioavailability of iron from beans can be inhibited by high concentrations of phytate and polyphenols [62]. The Lancet global diet comprises 50 g of legumes, 25 g of groundnuts, and 25 g of tree nuts per day [19]. Only three SSA countries meet this recommendation. Vanham et al. [24] have concerns regarding the recommended amounts of ground nuts and tree nuts, since they are very water-intensive, and hence are not affordable, sustainable, or climate friendly.

Per capita fruit and vegetable intake for most SSA countries falls far below the 500 g per day recommended by the WHO [46]. The healthy global diet recommends 300 g of vegetables (about 4 servings) per day and 200 g of fruit (about 2.5 servings) [19]. Only five SSA countries meet the Lancet global diet and the WHO recommendation for fruit and vegetables. Vegetables and fruit contribute substantially to fibre and micronutrients, and improve the dietary diversity of the diet. Additionally, there are many health benefits associated with a high intake of fruit and vegetables. These include a reduced risk of NCDs such as certain cancers, obesity, and type 2 diabetes [63]. Traditionally, in rural areas residents have vegetable gardens and fruit trees; however, with increased migration to cities, many people are now forced to spend money on fruit and vegetables. In addition, fruit and vegetables may be too often neglected in favour of less expensive refined cereals and highly processed foods [57].

The per capita intake of sugar in SSA countries is a serious consideration when examining elements of the nutrition and epidemiological transitions. Many SSA countries have a sugar intake greater than 60 g per day, with some having an intake greater than 100 g/day. The latter compare with the UK and USA, which have per capita sugar intakes of 105 g/day and 181 g/day, respectively. The WHO strongly recommends a free sugar intake of less than 10% of energy intake and a conditional recommendation of less than 5% of energy intake [64], while the Lancet global diet [19] recommends no more than 31 g of sweeteners per day. If one calculates the energy intake required by the average woman older than 18 years who is moderately active, a calorie intake of 2403 calories per day would be required [48]. A total of 10% of this, amounting to 240 kcal/day, comes from sugar, which equates to 60 g of sugar per day, placing many SSA countries above the WHO recommendation and most far above the Lancet global diet recommendation. The association between sugar and obesity has been well established. Obesity is a risk factor for numerous NCDs, including type 2 diabetes [65,66,67]. Data from 165 countries showed that diabetes prevalence was strongly correlated to per capita sugar consumption [68].

Furthermore, it has been shown that a high sugar intake may replace intake of micronutrients such as zinc, calcium, iron, and vitamins C, A, and B12 [69,70]. The average 330 mL serving of sugar-sweetened beverage (SSB) or fruit juice contains about 40–45 g of sugar (140–150 kcal) [71]. Due to mass advertising, popularity of sweet drinks, and lack of nutrition knowledge, SSBs may contribute considerably to daily calorie intake [68]. SSA appears to be an attractive market for beverage companies owing to its growing middle class and youthful populations.

The intake of meat and animal fat in most SSA countries is low, with only four countries possessing a per capita meat intake above 140 g/day, and the majority reaching less than 60 g per day. The Lancet global diet recommends 49 g of beef/lamb per week, 49 g of pork, and 203 g of chicken or other poultry [19]. If we add these meats, we come to 361 g per week, namely 66 g/day. The per capita intake of about half of the SSA countries lie below 45 g per day. Because of the high prevalence of anaemia in SSA [36], one does need to consider the importance of a small amount of animal flesh in the diet. However, at this time, for most SSA countries a high meat intake is not likely to contribute significantly to the nutrition transition, although one must recall that per capita food intake is not spread evenly among the population, and it is certainly expected that more affluent families would consume a higher amount of meat.

A high meat intake is associated with a Western diet [72], as shown by the high per capita intakes in the UK and USA, of 216 g/day and 352 g/day, respectively. Only three SSA countries have a per capita meat intake greater than 150 g/day. The diets of urbanised populations and populations from high-income countries are characterised by a higher content of meat, poultry, and other animal products than the diets of rural communities in low- and middle-income countries [73]. Meat fat comprises mostly monounsaturated fatty acids (MUFAs) and saturated fatty acids (SFAs). Trans-fatty acids comprise about 1–2% of total fatty acids across all types of meat; in ruminant meats they represent ∼2–4% [73]. Meat is an important source of high-quality protein and some essential micronutrients, such as iron, zinc, and vitamin B12. However, a high intake of red, and especially processed, meat is detrimental to health, as many epidemiological studies and associated meta-analyses have shown in the last decades. These detrimental effects include a higher risk of type 2 diabetes, cardiovascular diseases, weight gain, cancer, and stroke [73,74]. A recent review and meta-analysis showed an increased risk for type 2 diabetes and colorectal cancer, and for total mortality of cardiovascular diseases [74]. Only three SSA countries exceed the Lancet global diet recommendation for animal fat of 5 g/day.

Carvelho and colleagues used data from a population-based survey of 34,003 participants in Brazil whose usual meat intake was measured by 2 food records completed on 2 consecutive days [75]. The researchers’ aims were to determine the percent of the population who consumed more red meat than that recommended by the World Cancer Research Fund (WCRF) [76]—which recommends an intake of not more than 300 g of cooked meat per week (about 43 g/day)—and to examine the environmental impact of a high intake of red meat. They found that the mean intake of red and processed meat was 88 g/day, with more than 80% of the population consuming more than the WCRF recommendation. The entire Brazilian population contributed more than 191 million tons of CO_2_ equivalents, which could have been reduced to more than 131 million tons if the dietary recommendation was followed. In SSA countries, the median intake of meat in Clusters 1 and 3 are 42 g/day and 45 g/day, respectively, which would comply with the WCRF recommendation. Of particular importance is the effects of processed meat, which has been described as carcinogenic to humans [77]. In South Africa, for example, polony is one of the most common items consumed by children [78].

The majority of SSA countries also have a low intake of fish/seafood except for islands and countries with lakes. Most countries have a per capita fish/seafood intake of less than 30 g/day, which is less than the 28 g/day recommended by the Lancet global diet [19], and consequently have a low intake of omega-3 polyunsaturated fatty acids (PUFAs). Omega-3 polyunsaturated fatty acids (PUFAs) include α-linolenic acid (ALA; 18:3 ω-3), stearidonic acid (SDA; 18:4 ω-3), eicosapentaenoic acid (EPA; 20:5 ω-3), docosapentaenoic acid (DPA; 22:5 ω-3), and docosahexaenoic acid (DHA; 22:6 ω-3) [79]. Oils containing these fatty acids (FAs) originate primarily from certain plant sources, such as nuts and seeds, as well as marine sources. Long-chain (LC) ω-3 FAs such as EPA and DHA occur in the body lipids of fatty fish and the blubber of marine mammals [80].

Omega-3 PUFAs include numerous desirable health outcomes on prevention of cardiovascular diseases, atherosclerosis, thrombosis, inflammation, sudden cardiac death, cancer, diabetes, depression, age-related cognitive decline, rheumatoid arthritis, and periodontal disease. However, there are controversies with respect to the effect of ω-3 PUFAs on several health issues such as stroke, diabetes, cancer, and visual and neurological brain development [80]. According to a recent review by Elagizi et al., there is clear evidence from multiple studies that higher doses of omega-3 (2–4 g/day of EPA + DHA) appear to be safe and to reduce cardiovascular events in multiple populations. However, they do state that this warrants further study to conclusively determine the potential benefits of this well-tolerated, safe, and inexpensive therapy [79].

High intakes of inexpensive vegetable oils are also associated with the development of the nutrition transition [5]. However, per capita intake of vegetable oil is generally low in most SSA countries, lying below 30 g/day. There are four SSA countries in which intakes are similar to the UK and USA, e.g., more than 40 g/day. Many vegetable oils used in SSA are high in saturated fatty acids, e.g., palm and coconut oil, and many are high in omega-6 polyunsaturated fatty acids, such as sunflower and corn oil. The Lancet global diet [19] recommends a palm oil intake of no more than 6.8 g/day and unsaturated oils at 40 g/day. While many SSA countries appear to have a low intake of vegetable oil, fried foods are commonly eaten as snacks bought from street vendors [81,82]. These contribute significantly to energy intake and, in turn, promote obesity. A review by Dunbar et al. [83] stressed the finding that there is a shift in fatty acid profiles in high-income countries, with an increase in omega-6 polyunsaturated fatty acids relative to omega-3 fatty acids. This same shift is also emerging in low- and middle-income countries, and will lead to an increase in NCDs such as diabetes, CVD, and certain cancers.

The question is what the ideal diet is for SSA and, indeed, the rest of the world. Goldfarb and Sela [84] have produced a comprehensive review and meta-analyses of what this entails. They show that high meat consumption increases diabetes risk by 50% and mortality probability by an average of 18%. On the other hand, a plant-based, high-fibre diet decreases mortality by 15% and 20%, respectively, and diabetes risk by 27%. According to them, “the optimal diet is a whole food, high fiber, low-fat, 90+% plant-based diet.” Springmann et al. [85] project that by 2050, climate change will lead to per person reductions in global food availability, specifically in fruit and vegetable consumption. The health effects resulting from changes in dietary and weight-related risk factors due to climate change could be substantial.

According to the Lancet EAT Commission [19]: “Our universal healthy reference diet largely consists of vegetables, fruit, whole grains, legumes, nuts, and unsaturated oils, includes a low to moderate amount of seafood and poultry, and includes no or a low quantity of red meat, processed meat, added sugar, refined grains, and starchy vegetables. Our definition of sustainable food production stays within safe planetary boundaries for six environmental processes that together regulate the state of the Earth system, and include climate change, land-system change, freshwater use, biodiversity loss, and interference with the global nitrogen and phosphorus cycles. Applying a global food system modelling framework, we show that it is possible to feed a global population of nearly 10 billion people a healthy diet within food production boundaries by 2050. However, this Great Food Transformation will only be achieved through widespread, multisector, multilevel action that includes a substantial global shift towards healthy dietary patterns, large reductions in food loss and waste, and major improvements in food production practices.”

The present study has certain limitations. Food balance sheets include an estimate of food lost during distribution and processing, but do not take kitchen and plate waste into account. Furthermore, there are differences between mean data from food balance sheets and surveys which measure food consumption of individuals, with the latter generally providing better data. However, national surveys are expensive, and many countries cannot afford to carry them out on a regular basis. It is also important to note that mean per capita food values can be misleading because energy and nutrients are not distributed equally among the population. Thus, per capita values may indicate sufficient food security at the national level, but this does not identify groups or individuals who fall far below the mean values and others who greatly exceed them. Lastly, it is also important to keep in mind the constraints of the EAT diet in terms of affordability, sustainability, modelling, and availability of water [20,21,22,23,24].

## 5. Conclusions

Our description of the four clusters of SSA countries and a description of their climate as well as the Köppen climate classification have provided some insight into the diversity of climate in SSA and the food groups eaten in the various countries.

By superimposing clusters generated using macronutrient intake values on a map of the climatic regions in Africa, one can clearly see the importance of climate on the availability of food and food intake patterns in SSA. Climate change presents a big challenge to healthy eating, as the links between climatic regions and dietary intake are obvious. We examined the nutritional and epidemiological transition and associated development indicators of SSA countries. The conclusion is that SSA countries are at different stages of the nutrition and epidemiological transition, and it is a challenge to impose universal guidelines for healthy eating across SSA keeping in mind climate change, urbanisation, and a high birth rate as three major threats to food security.

When examining the per capita food intakes of SSA countries, we note many shortfalls in the diet. While many countries are still struggling with high levels of food insecurity and deficiencies such as iron-deficiency anaemia, numerous countries show indications of overnutrition and obesity. Probably the most important recommendations for a healthier diet would be to ensure that whole grains form a major part of the diet in preference to high intakes of refined cereals and starchy roots, and that fruit and vegetable intakes need to be considerably increased, as well as those of fish and seafood. Ideally, meat should be consumed mainly in the form of poultry and eggs, and processed meats should be avoided. Sweeteners should be reduced considerably, and preference should be granted to unsaturated oils in limited amounts.

## Figures and Tables

**Figure 1 ijerph-19-16791-f001:**
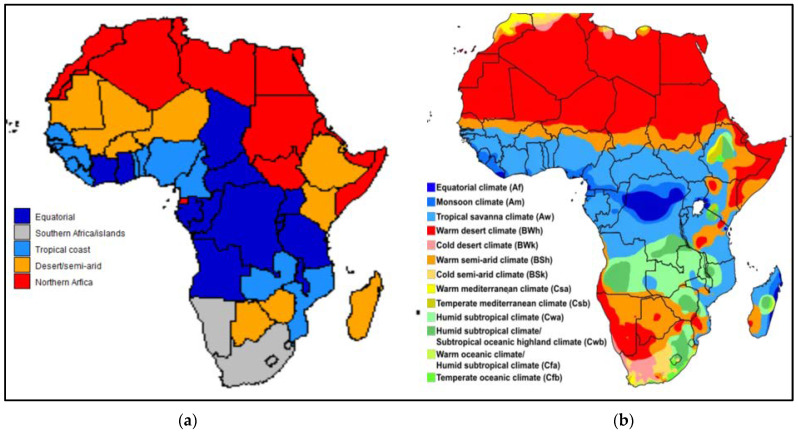
(**a**) Clustering of countries in SSA based on EFA and cluster analyses of food intake per day, measured in kcal/capita/day. Cluster 5 in this figure includes North African countries which were not used in the clustering process. (**b**) Africa map demonstrating the Köppen climate classification [44].

**Figure 2 ijerph-19-16791-f002:**
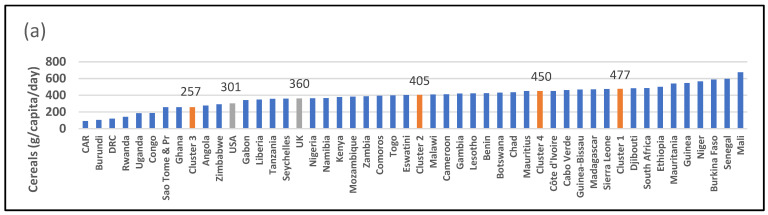
Data from food balance sheets for 2019 indicating gram per capita intake per day for (**a**) cereals, (**b**) starchy roots, (**c**) sugar and sweeteners, and (**d**) pulses and tree nuts. Red columns indicate medians of clusters; source: [25].

**Figure 3 ijerph-19-16791-f003:**
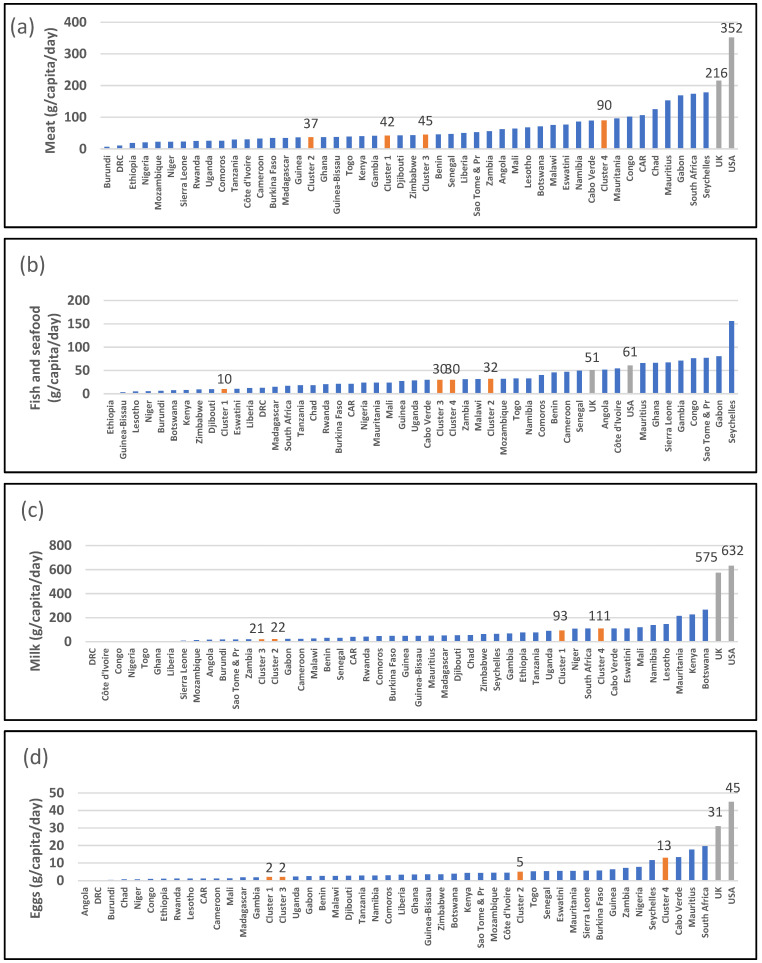
Data from food balance sheets for 2019 indicating gram per capita eaten per day for animal products. (**a**) meat, (**b**) fish/seafood, (**c**) milk, and (**d**) eggs. Red columns indicate medians of clusters; source: [25].

**Figure 4 ijerph-19-16791-f004:**
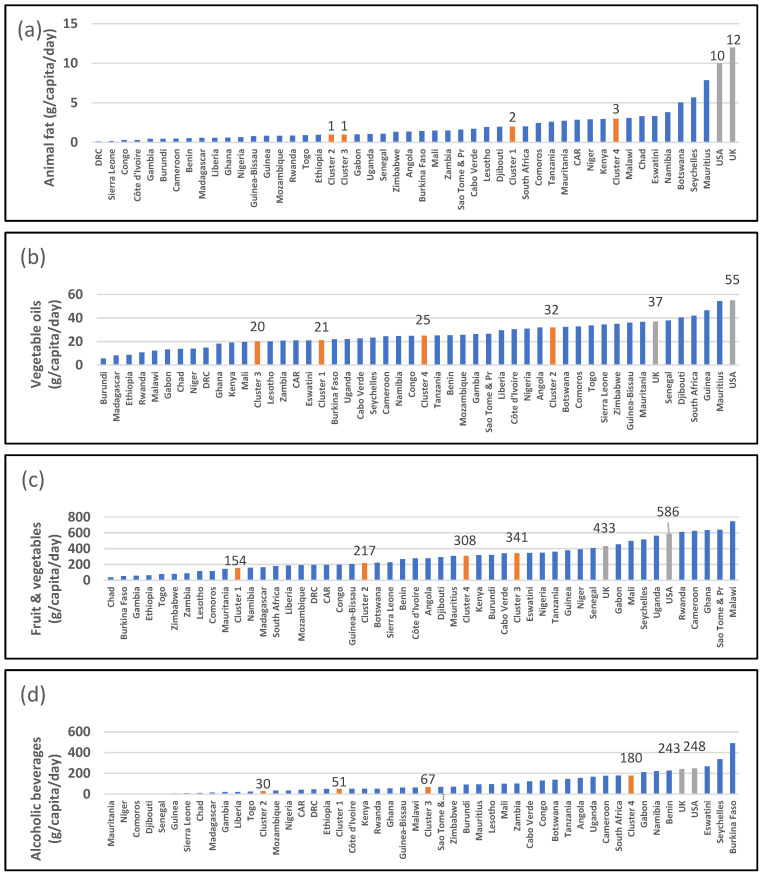
Data from food balance sheets for 2019 indicating per gram per capita intake per day for (**a**) vegetable oil, (**b**) animal fat, (**c**) fruit and vegetables, and (**d**) alcohol. Red columns indicate medians of clusters; source: [25].

**Figure 5 ijerph-19-16791-f005:**
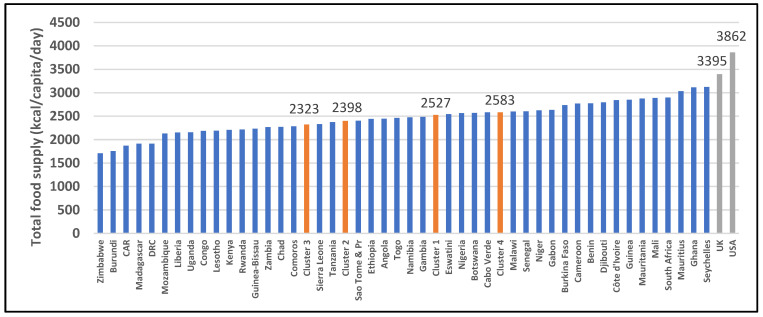
Data from food balance sheets for 2019 indicating per capita (kcal) of total food supply energy intake per day. Red columns indicate medians of clusters; source: [25].

**Figure 6 ijerph-19-16791-f006:**
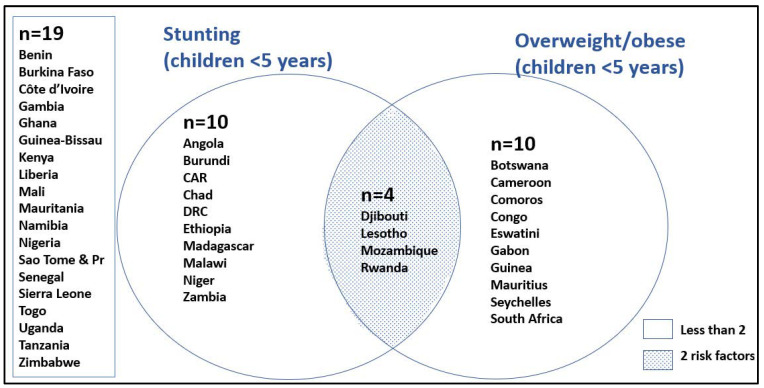
A Venn diagram showing SSA countries lying in the top tertiles for under five child stunting and child overweight (only those in the top tertile are in the diagram). Sao Tome & Pr: Sao Tome and Principe.

**Figure 7 ijerph-19-16791-f007:**
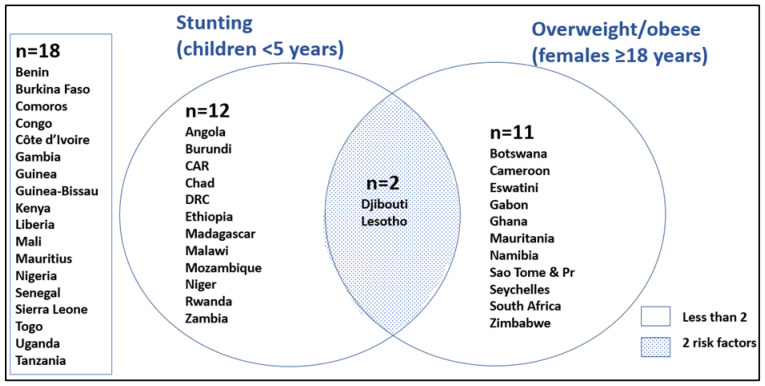
A Venn diagram showing the double burden of malnutrition in SSA countries illustrated by under-five child stunting and women overweight/obese (only those in the top tertile are in the diagram).

**Figure 8 ijerph-19-16791-f008:**
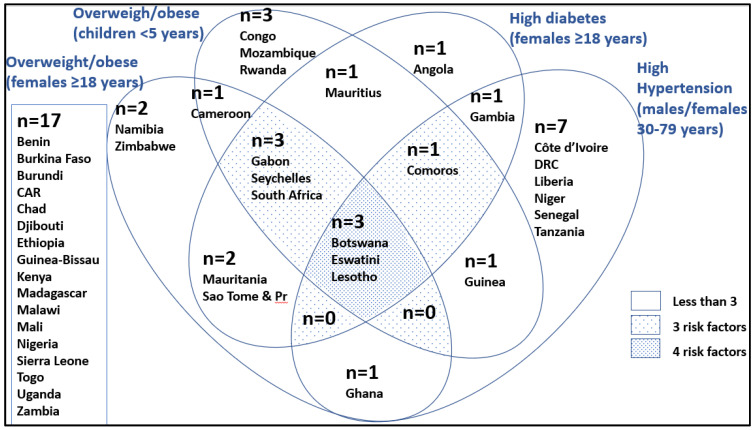
A Venn diagram showing evidence of the epidemiological transition in SSA countries (only those in the top tertile are in the diagram, i.e., 17 are not).

**Figure 9 ijerph-19-16791-f009:**
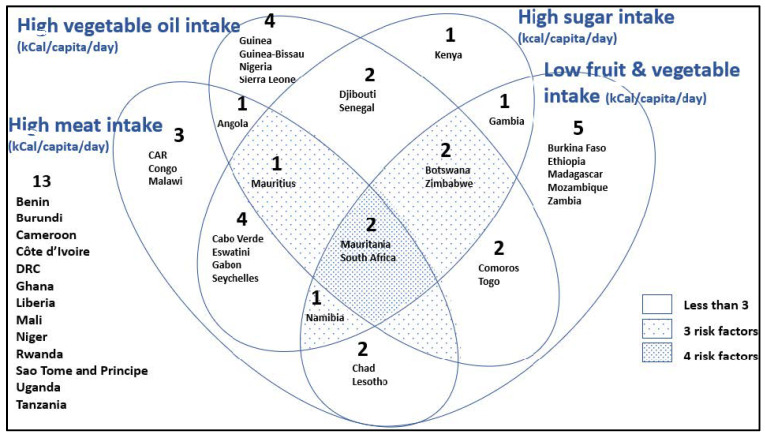
Venn diagram illustrating the nutrition transition in SSA countries. Those in highest tertile for meat, oil, and sugar intakes count as one risk factor each, while those with the lowest tertile for fruit and vegetables count as one. (only those in the top tertile are in the Venn circles of the diagram, i.e., 13 are not).

**Figure 10 ijerph-19-16791-f010:**
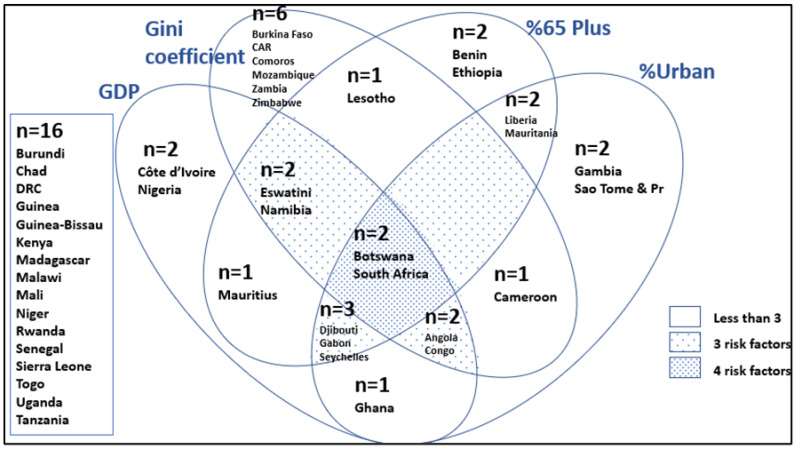
Venn diagram of development indicators for countries in SSA (only those in the top tertile are in the diagram, i.e., 16 are not). %65 Plus = people 65 years and older.

**Table 1 ijerph-19-16791-t001:** Food intake categories and subcategories as listed in the food balance sheets for 2019.

Code	Description
2905	Cereals excluding beer (wheat, rice, barley, maize, rye, oats, millet, sorghum, other)
2907	Starchy roots (cassava, potatoes, sweet potatoes, yams, other)
2909	Sugar and sweeteners (sugar, sweeteners, honey)
2911 + 2912	Pulses (beans, peas, other) and tree nuts
2913 *	Oil crops (soyabeans, groundnuts, sunflower seed, rape and mustard seed, cotton seed, coconuts, sesame seed, palm kernels, olives, other)
2914	Vegetable oils (soyabean oil, groundnut oil, sunflower seed oil, rape and mustard seed oil, cottonseed oil, palm kernel oil, palm oil, coconut oil, sesame seed oil, olive oil, maize germ oil, other)
2918 + 2019	Vegetables (tomatoes, onions, other), fruit excluding wine (oranges, lemons, grapefruit, other citrus, bananas, plantain, apples, pineapples, dates, grapes, other)
2922 + 2923 *	Stimulants (coffee, cacao, tea) and spices (pepper, pimento, cloves, other spices)
2924	Alcoholic beverages (wine, beer, fermented beverages, alcoholic beverages)
2943	Meat (bovine, mutton/goat, pork, poultry, other)
2945 *	Offal
2946	Animal fats (butter/ghee, cream, animal fat, fish body oil, fish liver oil)
2949	Eggs
2948	Milk excluding butter
2960	Fish, seafood (freshwater fish, demersal fish, pelagic fish, marine fish, crustaceans, cephalopods, molluscs, other)

Source: FAOSTAT [25,26]; * Not included in further analyses.

**Table 2 ijerph-19-16791-t002:** Summary of the factor loadings and the constructs after factor analysis on the per capita kcal/day intake of the 12 main food categories from the FAO food balance sheets.

	Factor 1		Factor 2		Factor 3	
	Food Item	Loading	Food Item	Loading	Food Item	Loading
Factor loadings > 0.3	Meat Animal fats Alcoholic beverages Fish and seafood Sugar/sweeteners Milk Eggs	0.74 0.69 0.68 0.51 0.50 0.43 * 0.44 *	Cereals Milk Animal fats Sugar/sweeteners	0.71 0.64 0.44 * 0.42 *	Vegetable oils Eggs Sugar/sweeteners Fish and seafood	0.76 0.53 0.43 * 0.32 *
Factor loadings <−0.4	-	-	Fruit and vegetables Starchy roots	−0.48 −0.83	Pulses and tree nuts	−0.53
% Variance explained (Total = 54.4%)	20.7%		20.6%		13.1%	
Type of diet	Westernised		Traditional/Westernised		Traditional/Westernised	

* The loading for this item is more than 0.3, although the maximum loading for this item is on another factor.

**Table 3 ijerph-19-16791-t003:** Description of the four clusters of SSA countries.

	Cluster 1 Desert/Semi-Arid N = 12	Cluster 2 Tropical Coastal N = 12	Cluster 3 Equatorial N = 14	Cluster 4 Southern African/Island N = 5
Countries	Botswana, Burkina Faso ^#^, Djibouti, Ethiopia ^#^, Gambia ^#^, Kenya ^#^, Lesotho *, Madagascar ^#^, Mali ^#^, Mauritania, Niger, Zimbabwe ^#^	Benin, Cameroon, Comoros, Guinea, Guinea-Bissau, Liberia, Mozambique^#^, Nigeria ^#^, Senegal ^#^, Sierra Leone, Togo, Zambia *	Angola, Burundi, Central African Republic (CAR), Chad ^#^, Republic of the Congo, Democratic Republic of the Congo (DRC) ^&^, Gabon, Ghana, Ivory Coast, Malawi, Rwanda, São Tomé and Príncipe, Uganda, Tanzania ^#^	Cabo Verde, Eswatini, Mauritius, Namibia, South Africa (Seychelles excluded as an outlier in the analysis)
Description	Warm desert and warm semi-arid climate.	Mostly coastal countries with tropical savanna and tropical monsoon climate.	Tropical savanna and subtropical climate areas in the equatorial region.	Mostly southern African countries and well-developed islands, with a spread of climate regions.
Exceptions which refer to countries above [33]	* Lesotho has cold semi-arid and tropical/sub-tropical regions. ^#^ Includes additional tropical/sub-tropical regions.	* Zambia has a humid subtropical climate. ^#^ Includes additional semi-arid regions.	^#^ Include additional semi-arid/desert regions. ^&^ Tropical rainforest.	South Africa: 5 major climate groups. Namibia: desert/semi-arid. Mauritius: Tropical. Eswatini: subtropical/semi-arid. Cabo Verde: desert.

Source of climate regions: [44,45].

**Table 4 ijerph-19-16791-t004:** Median and interquartile range g/capita/day values of the main food intake items of the food balance sheets, per cluster.

	Desert/Semi-Arid (Cluster 1) N = 12	Tropical Coastal (Cluster 2) N = 12	Equatorial (Cluster 3) N = 14	Southern African and Islands (Cluster 4) N = 5	Kruskal–Wallis *p*-Value	UK	USA
2901: All food items (kcal/capita/day)	2527 (2198–2764)	2398 (2249–2686)	2323 (2156–2600)	2583 (2544–2898)	0.211	3395	3862
2905: Cereals (g/capita/day)	477 [a] (422–552)	405 [a] (385–471)	257 [b] (141–356)	450 [a] (402–461)	0.0004 **	361	301
2907: Starchy roots (g/capita/day)	72 [c] (33–163)	400 [a] [b] (306–535)	736 [a] (431–796)	98 [b] [c] (94–186)	<0.001 ***	211	145
2909: Sugar and sweeteners (g/capita/day)	66 [b] (25–91)	33 [b] (26–45)	33 [b] (23–39)	110 [a] (97–134)	0.006 **	105	181
2911 + 2912: Pulses and tree nuts (g/capita/day)	31 (6–50)	28 (12–34)	23 (12–44)	22 (10–27)	0.870	12	18
2914: Vegetable oils (g/capita/day)	21 [a] [b] (16–34)	32 [a] [b] (25–35)	21 [b] (13–25)	25 [a] (23–42)	0.011 *	37	55
2918 + 2919: Fruit and vegetables (g/capita/day)	154 (71–305)	217 (152–365)	341 (197–611)	308 (180–341)	0.078	433	586
2924: Alcoholic beverages (g/capita/day)	51 (11–99)	30 (9–83)	67 (52–146)	180 (124–221)	0.022 *	243	248
2943: Meat (g/capita/day)	42 [b] (34–66)	37 [b] (25–47)	45 [b] (26–102)	90 [a] (86–153)	0.015 *	216	352
2946: Animal fats (g/capita/day)	2 [b] (1–3)	1 [b] (1–1)	1 [b] (0–3)	3 [a] (2–4)	0.006 **	12	10
2949: Eggs (g/capita/day)	2 [b] (1–4)	5 [b] (3–6)	2 [b] (1–3)	13 [a] (5–18)	0.001 **	31	45
2948: Dairy (g/capita/day)	93 [a] (59–181)	22 [b] (7–40)	21 [b] (5–42)	111 [a] (110–111)	<0.001 ***	575	632
2960: Fish and seafood (g/capita/day)	10 (6–22)	32 (26–46)	30 (19–66)	30 (17–31)	0.024 *	51	61

[a], [b], [c]: Different symbols indicate significant differences between cluster groups, Bonferroni, * *p* < 0.05, ** *p* < 0.01, *** *p* < 0.001

**Table 5 ijerph-19-16791-t005:** Median and interquartile range values of anthropometric and health data of Clusters 1–4.

	Desert/Semi-Arid (Cluster 1) N = 12	Tropical Coastal (Cluster 2) N = 12	Equatorial (Cluster 3) N = 14	Southern African and Islands (Cluster 4) N = 5	Kruskal–Wallis *p*-Value	United Kingdom	United States
Child stunting %HAZ < −2SD <5 years, 2021	27.7 (23.7–35.7)	30.1 (28.5–31.9)	31.8 (21.2–37.8)	22.1 (17.5–24.1)	0.1950	NA	NA
Child overweight %WHZ > +2SD < 5 years, 2021	2.1 [b] (1.5–5.4)	4.5 [b] (2.1–6.3)	3.4 [b] (2.3–4.5)	7.8 [a] (5.3–10.3)	0.0848	NA	NA
Concurrent stunting and overweight < 5 years, 2021	0.8 (0.5–1.8)	1.8 (0.7–3.5)	1.2 (0.6–1.6)	1.7 (1.1–2.6)	0.3398	NA	NA
Females 18 years and older, overweight BMI ≥ 25, age-standardised, 2017	37.0 [b] (29.6–48.7)	36.0 [b] (35.7–37.3)	34.9 [b] (31.5–39.5)	51.9 [a] (41.4–52.6)	0.3130	60.0	63.2
Child anaemia ^#^ <5 years, 2019 Hb < 110 g/L	52.1 [a] (43.3–72.0)	68.6 [a] (63.6–72.4)	58.8 [a] (56.1–62.4)	44.1 [b] (42.7–44.4)	0.0025 **	15.5	6.1
Women 15–49 years Anaemia ^#^, 2019 Hb < 110 g/L (pregnant) and Hb < 120 g/L (non-pregnant)	37.8 [a] (28.7–49.5)	48.0 [a] (41.6–50.6)	44.2 [a] (35.4–46.8)	25.2 [b] (24.3–30.5)	0.0098 **	11.1	11.8
Hypertension prevalence ^##^ in male and female, age-standardised 30–79 years, 2019, mmHg	36.7 (33.1–42.4)	36.7 (33.8–40.2)	35.6 (32.1–38.2)	38.1 (32.7–48.8)	0.8144	37.7	29.8
Females 18 years and older, type 2 diabetes ^###^, 2014, age- standardised	7.0 [b] (5.4–8.7)	6.9 [b] (6.4–7.2)	6.3 [b] (6.0–7.6)	11.3 [a] (8.0–12.6)	0.0291 *	4.9	6.4
Females’ total cholesterol, age-standardised, 18 years and older, 2018, mmol/L	4.1 (4.1–4.3)	4.2 (4.2–4.2)	4.1 (4.1–4.2)	4.2 (4.1–4.4)	0.5390	4.8	4.7
Females’ LDL-cholesterol, age-standardised, 18 years and older, 2018, mmol/L	2.9 (2.8–3.0)	2.9 (2.9–3.0)	2.9 (2.8–3.0)	2.9 (2.9–3.1)	0.8756	3.2	3.2
Females’ HDL-cholesterol, age-standardised, 18 years and older, 2018, mmol/L	1.2 [a] [b] (1.1–1.2)	1.1 [b] (1.1–1.2)	1.2 [a] [b] (1.1–1.2)	1.3 [a] (1.2–1.3)	0.0335 *	1.7	1.6

^#^ Anaemia: Percentage of women aged 15–49 years with a haemoglobin concentration less than 120 g/L for non-pregnant women and lactating women, and less than 110 g/L for pregnant women, adjusted for altitude and smoking; percentage of children aged 6–59 months with a haemoglobin concentration less than 110 g/L, adjusted for altitude [36]. ^##^ Prevalence of hypertension among adults aged 30–79 years, age-standardised. Hypertension is defined as SBP ≥ 140 mmHg, DBP ≥ 90 mmHg, or taking medication for hypertension [37]. ^###^ Diabetes is defined as fasting plasma glucose ≥ 7.0 mmol/L, or history of diagnosis with diabetes, or use of insulin or oral hypoglycaemic drugs [38]. [a], [b], Different symbols indicate significant differences between cluster groups, Bonferroni, * *p* < 0.05, ** *p* < 0.01.

**Table 6 ijerph-19-16791-t006:** Median and interquartile range values of some socioeconomic and demographic statistics for the four clusters.

	Desert/Semi-Arid (Cluster 1) N = 12	Tropical Coastal (Cluster 2) N = 12	Equatorial (Cluster 3) N = 14	Southern African and Islands (Cluster 4) N = 5	Kruskal–Wallis *p*-Value	United Kingdom	United States
GDP Current USD, 2020	996.0 [b] (784.0–1828.0)	1139.0 [b] (710.5–1416.0)	1086.0 [b] (710.0–2276.0)	5010.0 [a] (3916.0–6625.0)	0.0102 *	USD 2.8 million	USD 21.0 million
% Over 65 years (2020)	3.1 [b] (2.5–4.0)	2.9 [b] (2.8–3.1)	2.7 [b] (2.5–3.0)	4.8 [a] (4.0–5.5)	0.0040 **	18.7	16.6
% Urban dwellers	35.4 (28.5–59.0)	44.4 (40.0–50.2)	42.2 (23.5–66.8)	52.0 (40.8–66.7)	0.8288	83.9	82.7
% Informal urban dwellers	57.1 (46.5–64.3)	54.5 (50.1–69.4)	48.3 (42.1–65.1)	32.1 (25.6–42.3)	0.1369	-	-
Gini coefficient	41.2 [b] (36.0–46.1)	38.0 [b] (35.2–46.0)	41.4 [a] [b] (38.5–43.7)	54.6 [a] (42.4–59.1)	0.2046	35.1	41.5
Annual population growth (%) (2020)	2.6 [a] (1.8–2.9)	2.6 [a] (2.4–2.8)	2.5 [a] (2.4–3.0)	1.1 [b] (1.0–1.3)	0.0081 **	0.6	1.0
Birth rate (2020)	4.0 [a] (3.3–5.1)	4.5 [a] (4.2–4.7)	4.5 [a] (4.1–4.8)	2.4 [b] (2.2–2.9)	0.0043 **	1.6	1.6

[a], [b], Different symbols indicate significant differences between cluster groups, Bonferroni, * *p* < 0.05, ** *p* < 0.01

## Data Availability

Not applicable.

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
