# Peer review of "The Nutrition Transition and the Double Burden of Malnutrition in Sub-Saharan African Countries: How Do These Countries Compare with the Recommended LANCET COMMISSION Global Diet?"

_ijerph, 2022, doi:10.3390/ijerph192416791_

Round 1

Reviewer 1 Report

This paper covers an interesting topic. The title is strange. The paper does not explore what people in cities should eat based on availability and sustainability. In fact, the section of the paper that poorly addresses sustainable dies (but does not clarify the parameters) is very problematic. While the conclusions list what should be eaten, the justification does not come from the research but repeats (faulty) information from other sources that lack credibility and are presented without critique.

There is no definition of what a healthy diet should be based on the literature. In places, the paper assumes that the healthy planet diet is sustainable and the gold standard. It goes so far as to state (line 652) that this diet provides the framework for healthy eating and living. The EAT Lancet diet is not a guide to healthy living and will never be. It may not even be sound in its dietary assumptions.

There is no critical review of this. Only in the conclusions is the EAT Lancet diet identified as the ‘health planet diet’. Again, there is no thought or reflection on whether the EAT Lancet diet is actually healthy and sustainable. All such proposals are based on a set of beliefs and ideals, influenced by those involved in the steering committees and oversight groups of these processes. A large volume of literature has emerged since the publication of the EAT Lancet diet that challenges the assumptions and causality upon which the parameters have been set. There is no universal dietary guidance apart from the unquantified WHO guidance.

The term ulta-processed foods is defined on page 2 with no reference. It is important to use an officially approved definition. The examples are not appropriate and need to be sourced from credible peer-reviewed papers and authoritative sources. Why are only sweet biscuits listed when all commercially produced savoury and sweet biscuits are likely ultra-processed?  The statement that ‘these ulta-processed foods do not have the same nutritional benefits as unprocessed foods’ makes no sense as many ingredients of these foods have no comparisons in unprocessed forms.

Line 48 – over-nutrition is an incorrect term. Many overweight and obese people are malnourished in terms of mineral and vitamin deficiencies and sometimes also in protein. It is essential that terminology is used correctly. The authors should refer to the 2014 Rome Declaration on Nutrition for the correct terminology.

The terms traditional and typical westernised diets are not defined yet form the base of the classification of dietary patterns. This has to be corrected. See the HLPE report from 2020 for a classification. Using an internationally defined definition may change the results of this study.

So too, dietary transitions are not defined but diets are classified against this terminology.

It is unclear as to why the results are compared to the UK and USA? There is no justification for this. Using countries that are more similar to African nations would make far more sense.

The discussion of legumes in paragraph 2 (line 494 ff) does not consider the bioavailability of nutrients. This statement is biased by the unstated beliefs of the authors regarding the principles of the EAT Lancet diet. Who recommends that the ‘healthy planet’ diet contains 50g of legumes and 25g tree nuts per day? I assume the designers of the EAT Lancet diet? This is a typical point of the uncritical nature of this paper. The authors need to look at critiques such as that of Vanham et al (2020) (see https://doi.org/10.1016/j.gfs.2020.100357) who explain that:

·         Tree nuts and groundnuts are water-intensive products.

·         74% of irrigated nuts are produced under blue water stress.

·         63% of irrigated nuts are produced under severe water stress.

·         Increase in nut production needs to occur in a water-sustainable way.

Tree nuts are by no means planet-friendly foods!

It is not true that there are many health benefits associated with a high intake of fruit. Again, the authors are not being critical. Fruit and fructose are implicated in high insulin and insulin resistance and have detrimental effects on non-alcoholic fatty liver and other insulin-regulated health issues. This is contradicted in line 665.

From line 610 the authors set out to discuss what the ideal diet is for SSA. But this is simply a review of literature that should appear earlier in the paper. The sources used are biased and do not do justice to an enormous emerging body of literature that challenges the normative nutritional causation expounded in non-critical papers such as this. Springman’s work is the basis of the EAT Lancet report. Springman is not an authoritative nutritionist. Again, who says the EAT Lancet diet is actually a healthy diet?

The conclusions start with a statement that ‘one can clearly see the importance of climate on the availability of food and food intake patterns in SSA’. This is not clear from the data presented. The paper explored patterns of consumption and not climate-related production and availability.

There are far too many figures. These should be moved to an annex. It is unclear what the N is in figures 6 and 7 as the number of countries does not match the N on the left. There is a typo in Figure 8.

Below is a list of smaller challenges:

1.       Fruit does not have a plural in English. Both the plural and singular forms are simply fruit and not fruits. The two words are used interchangeably in the text.

2.       Line 163 – data on women’s – ‘s missing

3.       Line 74 provides a statistic for SSA’s ? (I assume GDP or turnover) for ? The sentence makes no sense. It has no reference source.

4.       Line 77 – what is ‘strategic urbanisation’?

5.       Line 92 – what is ‘improvement in government’?

6.       Line 175 – how was the data procured and form who? I suspect this means sourced?

7.       Figure 10 = what does %65 plus mean? People older than 65 years?

8.       Line 466 requires a reference.

9.       What is the relevance of the facts in line 470?

10.   Line 533 – why would sugar ‘dilute’ intake of micronutrients? I doubt this is possible. 

Author Response

I have added an attachment

Reviewer 2 Report

Authors collected large data from several international organizations’ databases and applied multivariate analyses to obtain interesting conclusions, which could be useful for researchers as well as policy makers in African countries. However, it is likely that the authors need to address the following possible mistakes/shortcomings. 1) “observed mainly in developing countries” (p.2) should be replaced with “observed mainly in developed countries.” 2) The authors used varimax rotation as the rotation method for factor analysis. As is well known, using varimax rotation assumes that there are no correlations between all factors. However, in the case of dietary intake, it seems to me that 'no correlation between each factor' is not a realistic assumption. A clear logic for the assumption that there is no correlation between each factor should be provided. 3) The Kruskal Wallis test provides an overall test statistic that enables to test the null hypothesis that all population medians are equal and show whether at least the median of one group is different from those of other groups. However, unless the authors compare the two groups by multiple comparisons, they may not be able to determine specifically which group is significantly different from the other group.

Round 2

Reviewer 2 Report

The authors estimated the factor scores using factor analysis based on oblimin and promax rotation and found that the maximum correlation between any two factors was less than 0.32. A correlation coefficient of 0.3 is usually considered a weak but significant relationship. However, the authors considered the correlations between factors were low and applied orthogonal varimax rotation. If any one of the correlation coefficients between each of the three factors are over 0.3, it seems to me that using varimax rotation is not justified. In my own field of study, we do the oblimin or promax rotation first, and if the correlation between the factors is weak, we use the varimax rotation after also logically explaining why it can be assumed that there is no correlation between the factors. Given that there is no logical explanation for no correlation between the factors, it is difficult to determine the extent to which the results of the analysis are valid.
